# Numerical Analysis of the Calcaneal Nail C-NAIL

František Sejda [1,2,*], Karel Frydrýšek [1,2,*], Leopold Pleva [1,3], Martin Pompach [4], Josef Hlinka [5],
Marek Sadílek [6], Zuzana Murčinková [7], Pavel Krpec [8], Miroslav Havlíček [9], Roman Madeja [1,3],
Jana Pometlová [1,3], Oldřich Učeň [10] and Kamila Dostálová [11]

1 Institute of Emergency Medicine, Faculty of Medicine, University of Ostrava, Syllabova 19, Vítkovice,
703 00 Ostrava, Czech Republic; leopold.pleva@fno.cz (L.P.); roman.madeja@vsb.cz (R.M.);
jana.pometlova@osu.cz (J.P.)
2 Department of Applied Mechanics, Faculty of Mechanical Engineering, VSB-Technical University of Ostrava,
17. listopadu 2172/15, 708 00 Ostrava, Czech Republic
3 Trauma Center, University Hospital Ostrava, 17. listopadu 1790, Poruba, 708 52 Ostrava, Czech Republic
4 Nemocnice Pardubického Kraje, Kyjevská 44, 53 203 Pardubice, Czech Republic; martin.pompach@nempk.cz
5 Department of Material Engineering, VSB-Technical University of Ostrava, 17. listopadu 15/2172, Poruba,
708 00 Ostrava, Czech Republic; josef.hlinka@vsb.cz
6 Department of Material Technology, Faculty of Mechanical Engineering, VSB-Technical University of Ostrava,
17. listopadu 2172/15, 708 00 Ostrava, Czech Republic; marek.sadilek@vsb.cz
7 Faculty of Manufacturing, Technical University of Košice, 080 01 Prešov, Slovakia;
zuzana.murcinkova@tuke.sk
8 V-NASS, a.s., Halasova 2938/1a, Vítkovice, 703 00 Ostrava, Czech Republic; pavel.krpec@v-nass.cz
9 Medin, a.s, Vlachovicka 619, 592 31 Nové Město na Moravě, Czech Republic; miroslav.havlicek@medin.cz
10 Department of Machine and Industrial Design, Faculty of Mechanical Engineering, VSB-Technical University
of Ostrava, 17. listopadu 2172/15, 708 00 Ostrava, Czech Republic; oldrich.ucen@vsb.cz
11 Centre of Advanced Innovation Technologies, VSB-Technical University of Ostrava, 17. listopadu 2172/15,
708 00 Ostrava, Czech Republic; kamila.dostalova@vsb.cz
* Correspondence: frantisek.sejda@hella.com (F.S.); karel.frydrysek@vsb.cz (K.F.); Tel.: +420-737914234 (F.S.)

**Abstract:** The presented article investigates the biomechanics of the calcaneal nail C-NAIL[TM] by
numerical calculations and, partially, experimentally. This nail is widely used in trauma and or-
thopaedics. A numerical model of implants directly interacting with the bone tissue model obtained
from CT scans was calculated. The material properties of the bone tissue can be described by several
models; in this work, a non-homogeneous material model with isotropic elements and prescribed
elastic modulus was used to provide a more accurate model of the applied force distribution on the
individual parts of the implants. The critical areas of the nail and its fixtures were investigated using
finite element strength calculations to verify their strength and reliability, contributing to the safety
and faster and easier treatment of patients. These analyses suggest that the strength of the calcaneal
nail C-NAIL, as well as the stabilization of bone fragments resulting from its use, are sufficient for
clinical practice.

**Keywords:** traumatology; orthopaedics; calcaneus C-NAIL; osteosynthesis; biomechanics; finite

## 1. Introduction

The calcaneus or heel bone (Figure 1) is the biggest bone in the human foot, giving it
its shape and functionality. In the foot skeleton, the heel bone forms the rear and bottom
parts that bear the body weight through the talus bone. Therefore, the upper part of the
heel bone joins the talus bone and forms a part of the ankle joint (talocrural joint) (Figure 2);
the front part is adjacent to the cuboid bone (Figure 2) and in the rear part, the Achilles
tendon attaches to it [1,2].

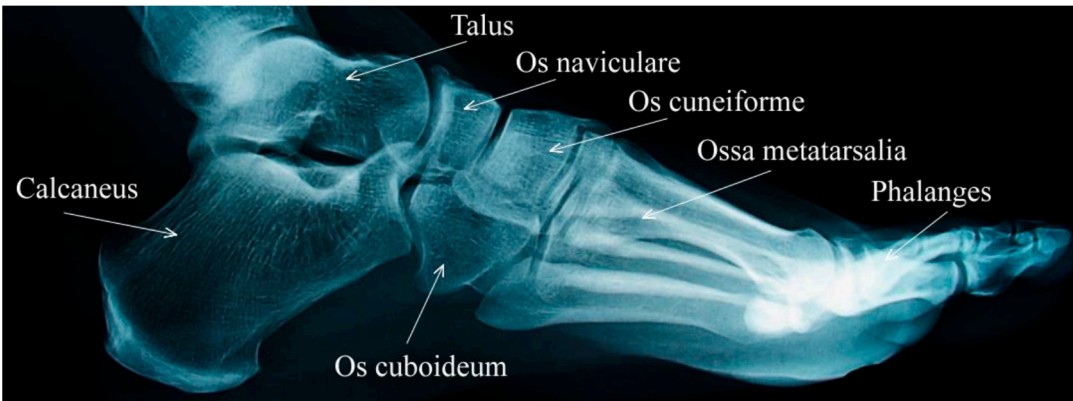

**Figure 1.** X-ray of the foot.

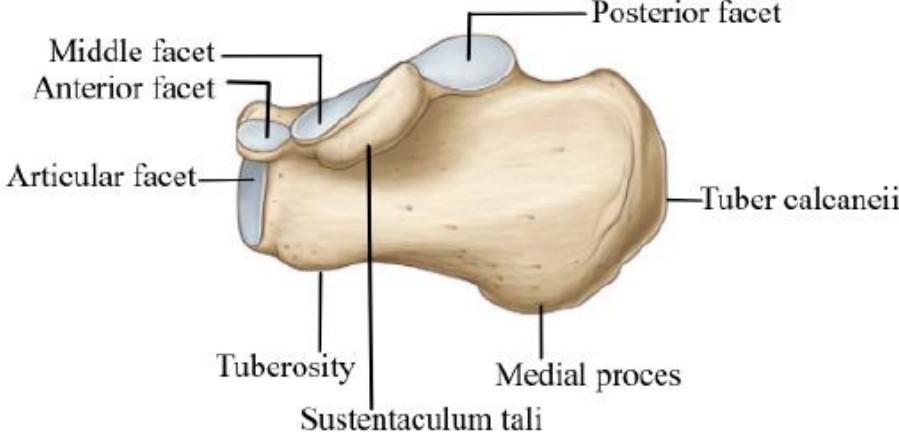

**Figure 2.** Anatomy–joint surfaces of the heel bone.

A dislocated heel bone fracture counts among the most complicated injuries of the lower limb and, usually, is associated with permanent damage. Such a fracture also represents a major trauma for the patient. Patients have great difficulty walking, and the damage can lead to the development of arthrosis and pain; in most cases, it is impossible to walk at all. Therefore, osteosynthesis is often used as the treatment of choice. In addition, the bone is directly under the skin and is not covered by muscles but rather only by a fat pad, which (in the case of damage to the superficial structures) leads to additional complications in the healing of both the skin cover and the bone itself [2,3].

The first references to the treatment of calcaneal fractures date back to the time of Hippocrates (460–385 BC). The first recorded resting-based treatment leading to the stabilization of the bone fragments comes from French Petit and DeSault in 1720. In one of the earliest editions of the American medical journal (1880), Bailey described the treatment by rest, bandage, and saline solution. A major breakthrough in the treatment came with the discovery of X-rays, when the understanding of these fractures fundamentally changed and the first efforts at anatomical repositioning appeared. In 1913, the French physician Lerich performed the first osteosynthesis using a splint with screws and bone grafts. Throughout the 20th century, the development of new methods and approaches for open repositioning and internal fixation continued but for most surgeons, conservative treatment (administration of solutions and analgesics, immobilizing casts) still represented the method of choice. The truly massive use of internal and external fixation begun in the 1980s. Together with the development and widespread use of CT (computed tomography) scanning, more accurate diagnosis and classification of these fractures (e.g., according to Sanders) was made possible [2–4].

Today, open repositioning combined with internal fixation using a calcaneal plate is the standard method of treatment; however, it is occasionally complicated by problems

with healing of the surgical wound and possible infection. For this reason, new approaches to internal fixation (e.g., mini-invasive approach using nails that do not need such an extensive skin cover opening during insertion into the fracture, thus minimizing the risk of infection) have been developed. The success of heel bone treatment also depends on the strength of the used plates and calcaneal nails; in case of their failure, it is necessary to perform a new osteosynthesis. Despite the best efforts of the surgeons, the success rate of the treatment is not 100% (e.g., infection, osteolysis, etc.); however, modifications and improvements of these implants have led to a reduction in the probability of their failure [2–4].

Widely accepted methods of internal fixation include the method of fixing the plate with screws directly to the bone (Figures 3 and 4). This method is used for fractures of the body of the bone. However, fractures often occur in the neck. In such cases, other methods are used, such as nailing, i.e., driving a nail into the bone cavity. Leading manufacturers of external and internal fixators for the heel bone include, for example, the Turkish Normmed, see [5], which specializes in the treatment of small bones, the Swiss Medartis [6], GPC Medical Ltd., see [7], the Czech manufacturer Medin a.s. [8], etc.

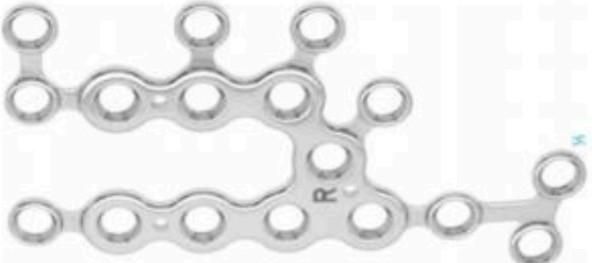

**Figure 3.** Internal fixator–plate for treatment of heel bone fracture (alternative method of treatment).

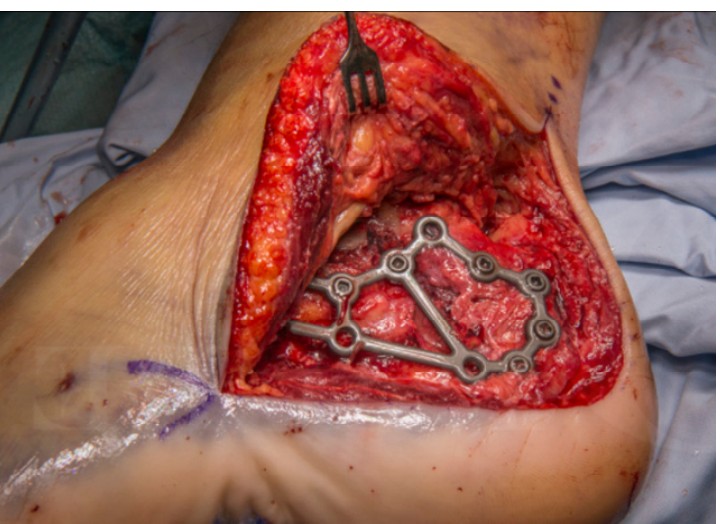

**Figure 4.** Lateral incision and plate application during heel bone osteosynthesis (alternative method of treatment).

The latter company, Medin a.s. [8], developed a calcaneal nail for angular stabilization under the commercial name C-NAIL (Figure 5). This calcaneal intraosseous nail is used for the mini-invasive fixation of intraarticular and extraarticular fractures of the heel bone, stabilizing the fragments of the heel bone by the nail in conjunction with seven locking screws; this leads to the formation of angularly stable fixation. To achieve maximum stability, the sustentacular fragment (i.e., part or whole broken sustentaculum tali) is fixed into the nail with two locking screws guided by a targeting device [8]. Mini-invasiveness is one of the major advantages of using C-NAIL. A small lateral approach of about 3 cm from

the apex of the outer ankle towards the base of the 5th metatarsal is sufficient for fragment repositioning; further, just a few mini-incisions are needed for introducing Kirschner guide wires, the actual nail, and the individual screws. This mini-invasiveness significantly reduces the risk of possible infection. The stability of the implant and, thus, the firm fixation of the fragments constitute additional advantages.

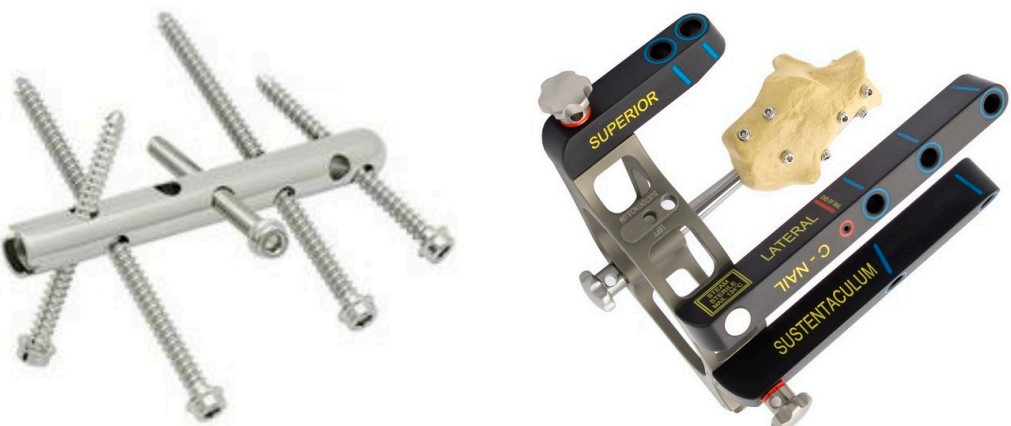

**Figure 5.** Internal fixator C-NAIL for treatment of the heel bone fractures and the targeting device.

The lower age limit for using this treatment is given by the closure of the growth cleft of the child's heel bone, which closes at the age of 16–17 years. The upper age limit is not specified, it depends on the biological condition of the patient. From doctoral experiences: the youngest patient was 17 years old, and the oldest patient was 75 years old.

Presently, two types of materials are compatible with human tissue (titanium alloy Ti6Al4V and surgical stainless steel AISI 316L) are most widely used for osteosynthetic plates and nails [9], and locking screws are also from the same materials. The diameter of the screws is 3.5 mm, but the length varies from 22 mm to 70 mm [8]. Here, the finite element (FE) analysis of a calcaneal nail made of titanium alloy will be performed. A homogeneous isotropic linear elastic material model will be used for the actual analysis. The material constants were determined by tensile testing [10]. The modulus of elasticity E was determined to be 105 666 MPa, and the Poisson's number, which indicates the ratio between longitudinal and transverse elongation, for this titanium is alloy $\mu_{titanium}$ = 0.342, see [11].

## 2. Numerical Bone Modelling

The acquisition of a series of consecutive images forms the basis for creating a correct geometrical and numerical bone model. CT (computed tomography) or MRI (magnetic resonance imaging) are the methods most commonly used for imaging internal organs and bones. For soft tissue imaging, MRI imagery is preferable because of the use of magnetic fields and high-frequency electromagnetic waves (i.e., zero radiation load for both the doctor and the patient) [12].

CT images of the lower limb were used to create good quality models of the heel bone (Figure 6). The Materialise Mimics software [13] was used to subsequently obtain a CAD model of the calcaneus. The proper calculation of the bone model needs to separate the dense bone tissue from other soft tissues in the individual CT sections. The Mimics software performs this separation based on the Hounsfield units (*HU*), which is the density of individual pixels.

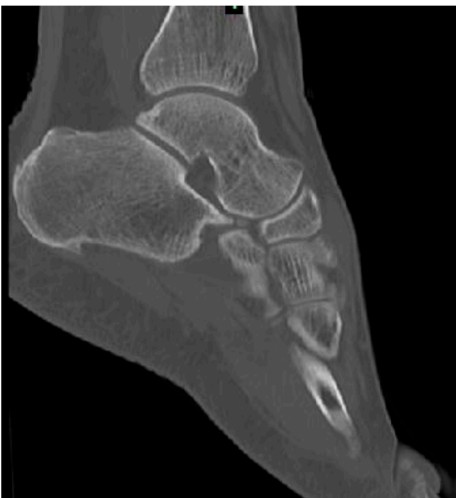

**Figure 6.** CT image of the foot–sagittal plane.

*HU* can be perceived as the level of grey on the X-ray. CT images can be subsequently used to create a 3D model of a bone, organ, or soft tissue. The difficulty of this process largely depends on the quality of the CT imagery—where the CT quality is low, the soft tissues must be distinguished from the bone manually. The CAD model of the heel bone presented in Figure 7 was prepared from CT images provided by the University Hospital Ostrava, Czech Republic. This model was subsequently subjected to FE analysis and served as a basis for the stability analyses of calcaneal nails.

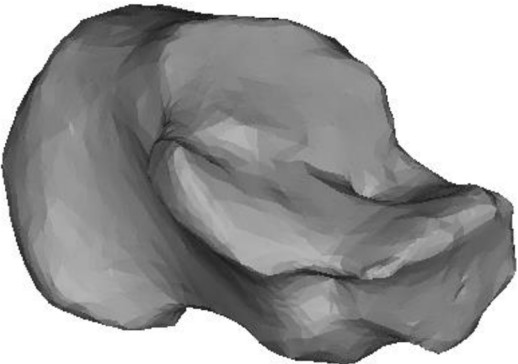

**Figure 7.** The anatomical CAD model created from CT imagery in the Mimics software.

## 3. Material Properties of the Bone

Human bone tissue is a non-homogeneous and anisotropic material. The mechanical properties of bone tissue, which are genetically determined, are therefore strongly dependent on the direction of loading, age, sex and health of the person. Under compressive loading, the ultimate strength is higher than under bending loading [14]. Therefore, the search is on for suitable models describing the complicated mechanical properties of bone tissue that are as close to reality as possible. According to the literature, the use of a non-homogeneous isotropic material model described by the elastic modulus E and the Poisson number μ for each element of the finite element mesh seems to be a sufficient approximation to the real bone.

The magnitudes of the elastic modulus are calculated from the magnitude of *HU* in the tissue volume. The density of the zone of interest according to *HU* can be further determined from the CT images. Using the Materialise Mimics software and mathematical and physical relationships, material properties can be assigned to each element. There is a large body of literature and scientific articles dealing with the conversion of Hounsfield units to densities ρ and elastic moduli E, see [14–19] and others. The mechanical properties

of bone tissue vary depending on gender, age, health status (smoker, diabetic), tissue type, genetics, etc. A method discussed in the literature [18] was used for recalculation of the density and the elasticity modulus (Figure 8).

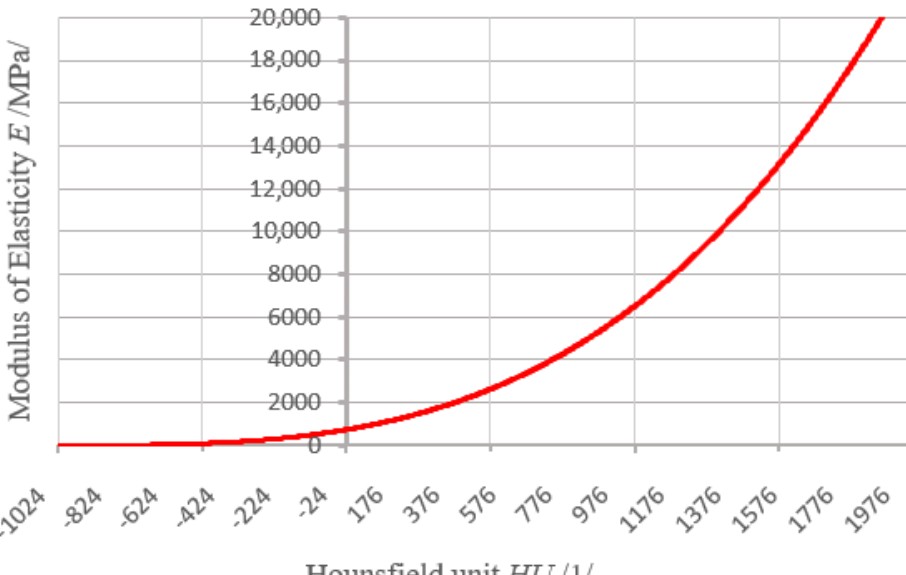

**Figure 8.** Association of the elasticity modulus and Hounsfield units.

This approximation of reality is sufficient for the presented research as it primarily focuses on the analysis and optimization of the calcaneal nail; in this paper, therefore, the bone itself is not in the spotlight of interest here.

As a function, The calcaneal density $\rho = \rho_{(x, y, z)}/\text{kg·m}^{-3}/$ can be expressed as:

$$\rho = x + HU \times y \tag{1}$$

where $x$, $y$ and $z$ are coordinate axes with a suitably selected origin of the coordinate system. Values of $HU = HU_{(x, y, z)}$ are acquired via CT snapshots.

Similarly, the elasticity modulus $E = E_{(x, y, z)}$ can be expressed by constants $A$, $B$ as:

$$E = A \times \rho^{B}_{(x,y,z)} = 9354 \times 10^{-7} \times \rho^{3.15}_{(x,y,z)} \tag{2}$$

The distribution of the elasticity modulus is presented in Figures 9 and 10.

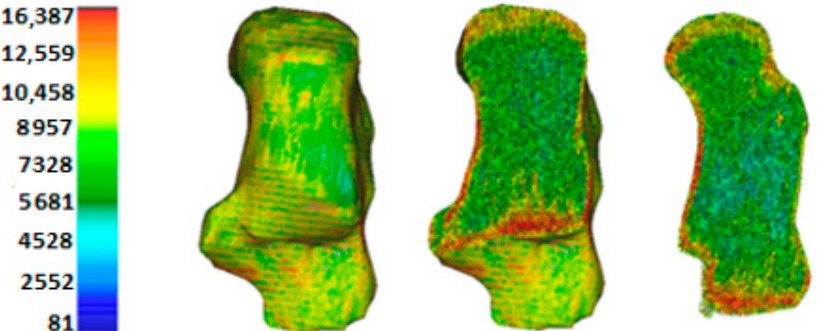

**Figure 9.** Distribution of the elasticity modulus $E/\text{MPa}/$for the heel bone (100 materials acquired from CT images via Mimics software)–transversal projection.

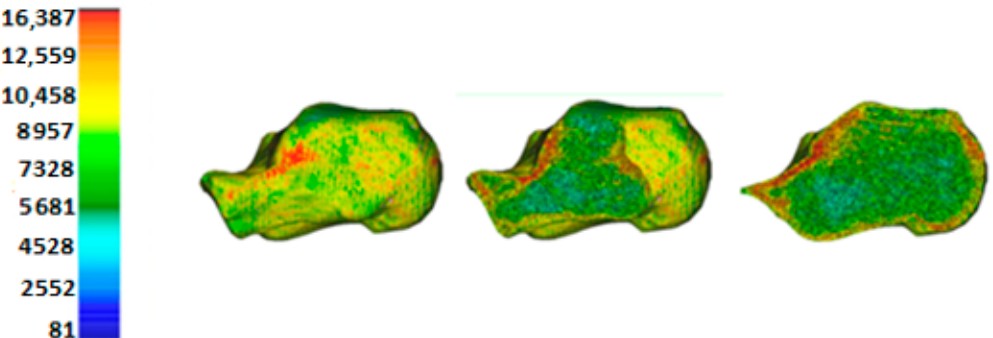

**Figure 10.** Distribution of the elasticity modulus $E/\mathrm{MPa}/$for the heel bone (100 materials acquired from CT images via Mimics software)–sagittal projection.

### 3.1. Strength Analysis of a Healthy Calcaneus without a C-NAIL

Before the actual FEM calculations of the osteosynthetic implants, a stress–strain analysis of the whole (healthy) heel bone was performed as below.

The computational model was based on a test machine constructed for static and dynamic (fatigue) experiments described in [20], where the anterior part of the bone was supported by a substitute of the cuboid bone (Figure 11), and the load was applied by a hydraulic test machine through the substitute of the tibial joint to simulate the normal loading of the foot. For the purpose of the experiment, these cuboid and tibial replacements were made of veterinary bone cement. This method (i.e., the replacement of the surrounding skeleton with veterinary or dental cement) has been previously described [21–23] and therefore will not be described in greater detail here.

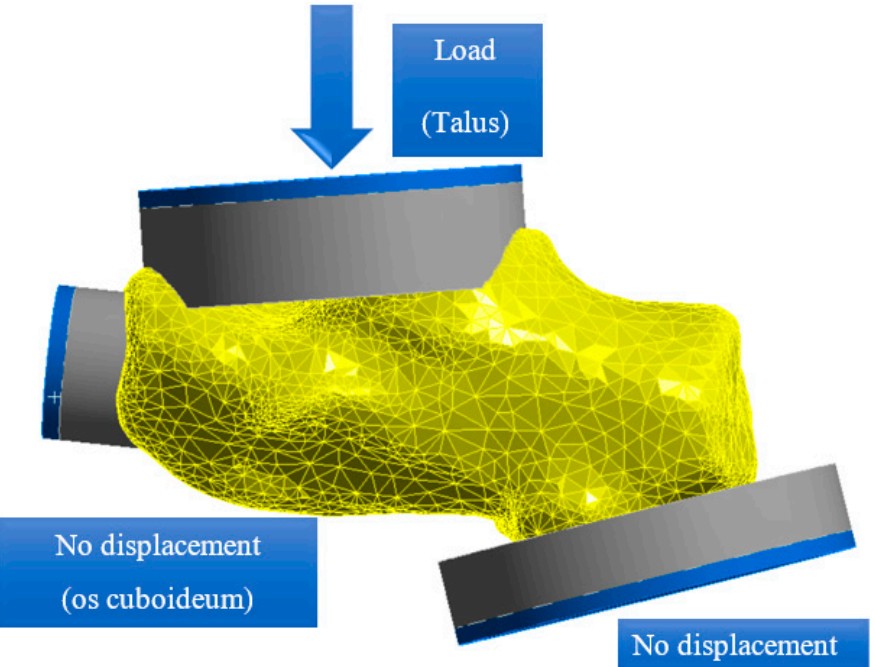

**Figure 11.** Boundary conditions.

The computational model (Figure 11), similar to an experiment, replaces the interaction of the heel bone with the other leg bones by finite elements with the mechanical properties of cured epoxy putty [24]. This proxy (epoxy putty instead of veterinary cement, the mechanical properties of which were unavailable) was selected on the basis of reasonable expert estimates of the mechanical properties of the modulus of elasticity of the used veterinary bone cement (DEMOTEC95-Demotec, Nidderau, Germany), $E = 4830$ MPa, $\mu_{demotec} = 0.3$.

Individual parts of the hydraulic testing machine are modelled as elements with mechanical properties of common structural steel, $E$ = 210,000 MPa, $\mu$ = 0.3. FEM analysis was performed for a model of the heel bone created from CT images, with a material model respecting the mechanical properties of a real bone described element by element by the gradually changing modulus of elasticity according to Equation (2). In this experiment, the upper (proximal) solid elements representing the talus/upper part of the ankle joint were loaded with an axial force, see Equations (3) and (4).

$$G = m \cdot g = 120 \cdot 9.81 = 1177.2 \text{ N} \tag{3}$$

$$F = Kdyn \cdot G = 1.47 \cdot 1177.2 = 1722.84 \text{ N} \tag{4}$$

where $G/N/$is the static gravitational force, $F/N/$is the dynamic force, and $Kdyn$ is the dynamic coefficient converting the static problem to a quasi-static problem; $g$ = 9.81/m s$^{-2}$/is the gravitational acceleration.

To calculate the healthy (solid) calcaneus using FE analysis, a patient with a mass of $m$ = 120 kg was considered. Calculations were performed for the maximum dynamic value that can be transferred to the heel bone, namely the total patient's dynamic force $F$. In addition, a dynamic coefficient $Kdyn$ = 1.47 was introduced as a simple solution to converting the static problem into a dynamic one (a common engineering approach; the $Kdyn$ value was established based on our previous measurements of the dynamic load of the foot). The loading force is therefore equal to the product of the dynamic coefficient $Kdyn$ and the patient's gravitational force $G$.

The elements representing the cuboid bone and the substrate are tightly constrained at all degrees of freedom on the free end faces (highlighted in blue; Figure 11).

### 3.2. Strength Analysis of a Healthy Heel Bone without C-NAIL

The distribution of reduced stress under the HMH (von Mises stress) hypothesis for the solid (healthy) calcaneus generated from CT images and described by a variable linear homogeneous isotropic material model (distribution of 100 material properties across the elements in the calcaneus) is shown in (Figure 12). The largest value of the reduced stress is 11.9 MPa, which is still relatively low stress, located at the point of the contact of the bone model with the veterinary cement (Figure 11).

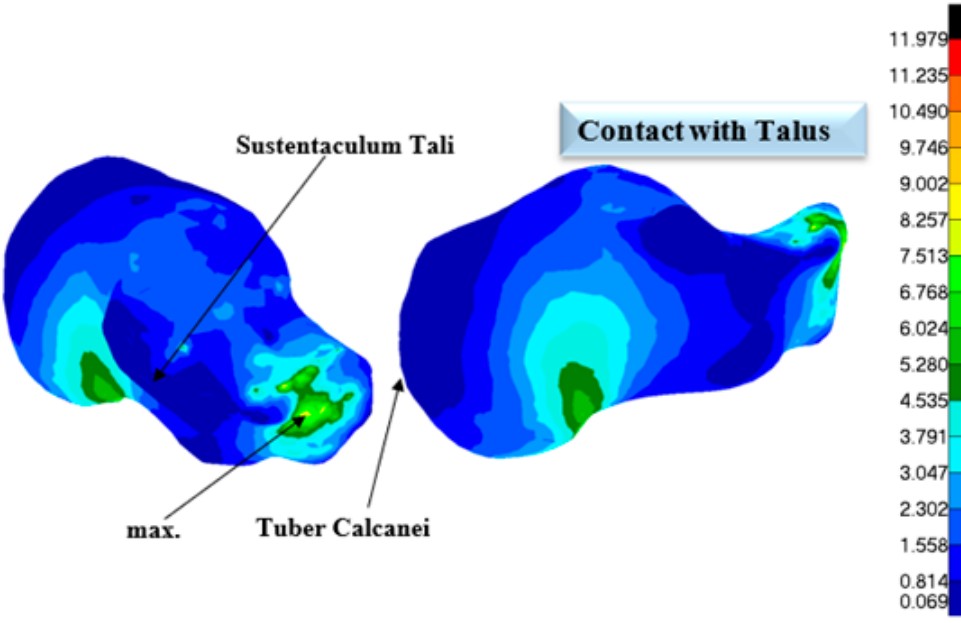

**Figure 12.** Distribution of the reduced stress on the healthy heel bone produced from CT images, according to the HMH/MPa/hypothesis.

Figure 13 shows the distribution of total displacements for the healthy heel bone under stress created from CT images as described by a variable linear homogeneous isotropic material model (100 materials). The greatest total displacement of 0.44 mm was detected in the sustentaculum tali region.

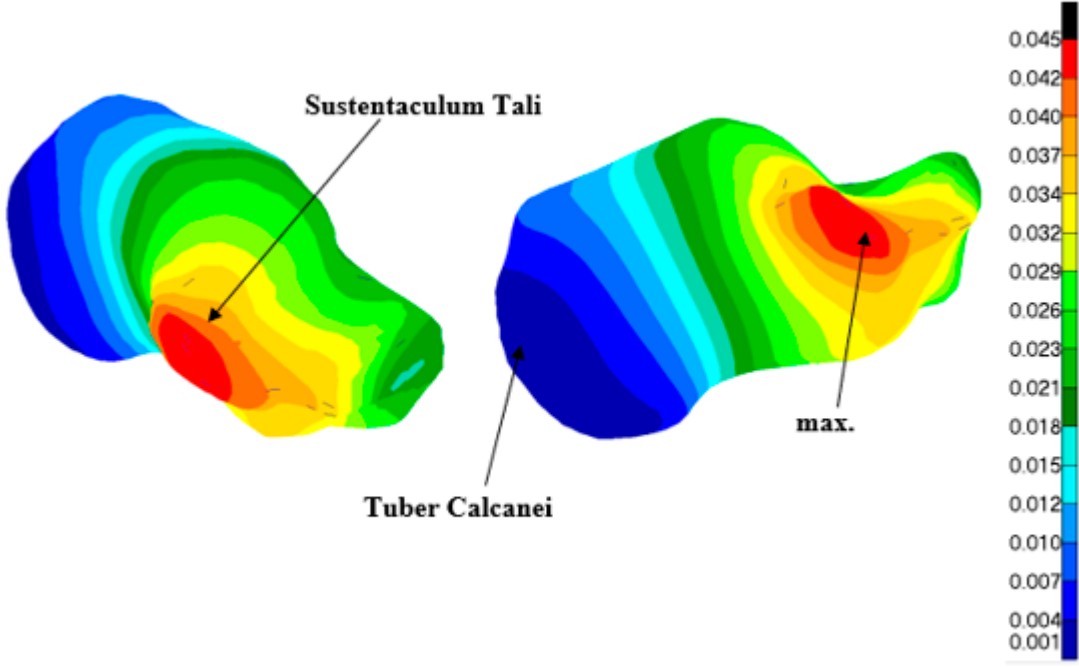

**Figure 13.** The total displacements on healthy calcaneum created from CT imagery /mm/.

The areas with greater values of the HMH stress are those that are most susceptible to fracture (Figure 14; lines between points A-B, B-C). The resulting HMH stress distributions were presented to medical experts who confirmed that these fracture lines correspond to the typical fracture lines on a real bone.

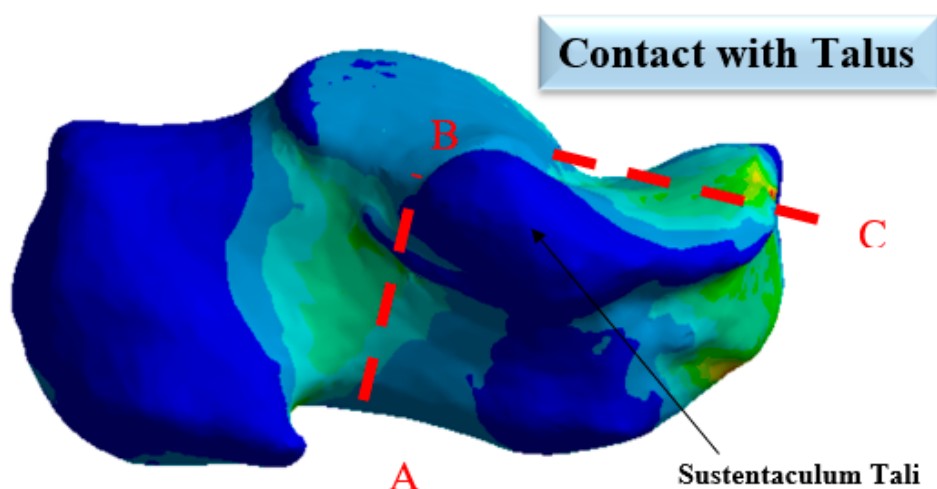

**Figure 14.** The most likely fracture lines of a healthy heel bone.

## 4. Strength Analysis of the Calcaneal Nail C-NAIL in Interaction with the Heel Bone

### 4.1. Numerical Model

The strength analysis of the C-NAIL calcaneal nail was performed on models of the heel bone virtually "cut" into seven fragments corresponding to complicated comminuted fractures (Figures 15 and 16) according to the Sanders IIB classification [25], similar to the experiment in [23].

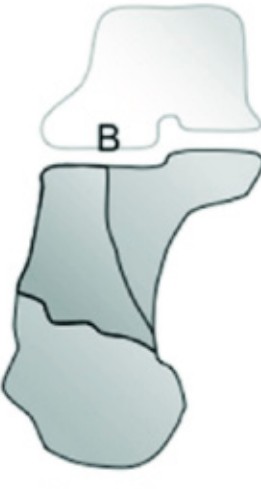

**Figure 15.** Sanders classification of calcaneal fractures, Type IIB—adapted with permission from Ref. [24].

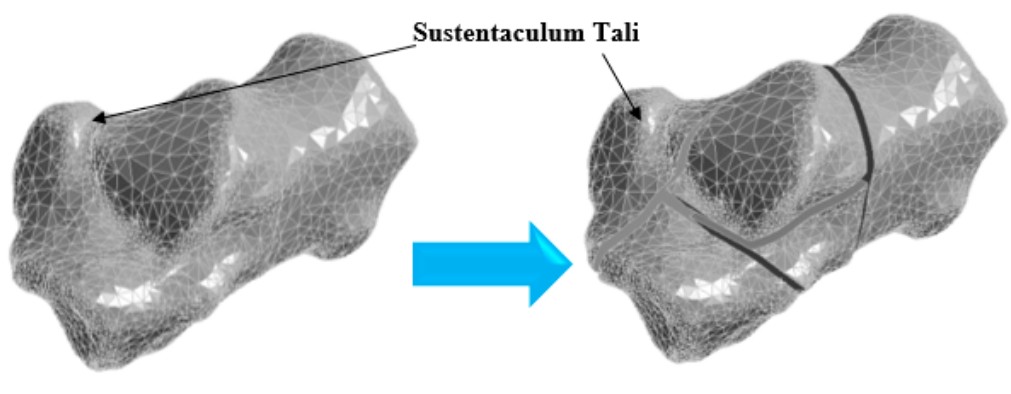

**Figure 16.** The adjustment of the bone model for calculations.

Further model parameters, including the material and loading, are presented in Section 3.1.

### 4.2. Calculation Results

The maximum reduced stress calculated according to the HMH theory (about 390 MPa) is exerted on the screw passing through the fragments of the heel bone (Figure 17; the fragment of the calcaneus with sustentaculum tali, Figure 1). The upper central fragment (the fragment with the posterior facet, Figure 1) of the heel bone tends to move downward, causing increased stress. Thus, the majority of the load is transferred to the C-NAIL, due to the lack of adhesion between the heel bone fragments, which is the desired result.

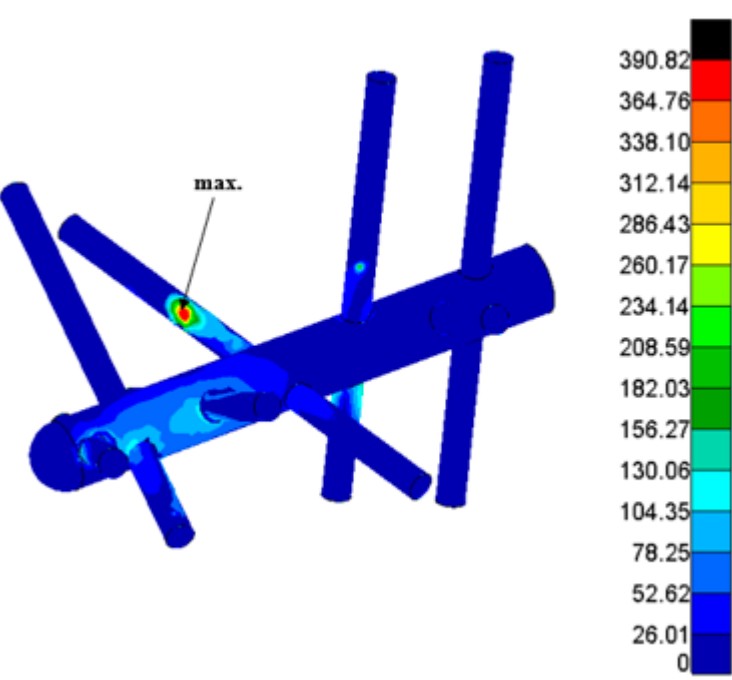

**Figure 17.** Distribution of the reduced stress on the calcaneal nail according to the HMH hypothesis /MPa/.

The stress on the heel bone itself, approx. 72.8 MPa (Figure 18), is negligible compared to the nail; the maximum stress is detected at the site of contact of the bone with the veterinary cement (Figure 18).

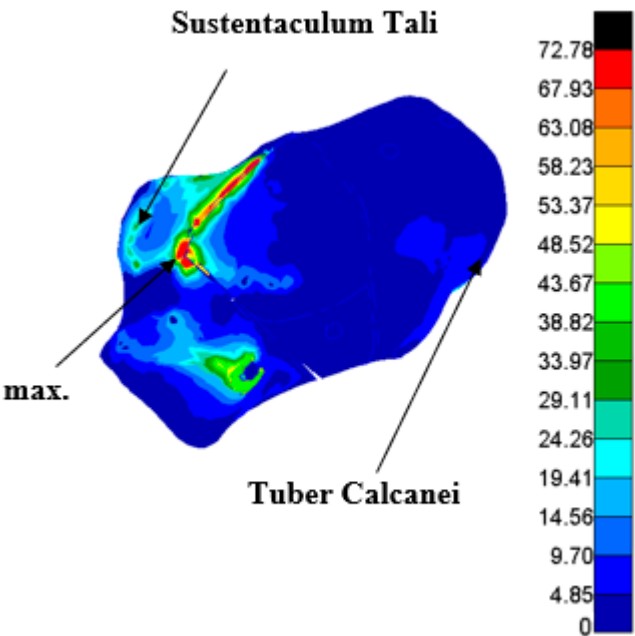

**Figure 18.** Distribution of the reduced stress on the fractured heel bone containing C-NAIL according to the HMH theory /MPa/.

The distribution of displacements on the heel bone is shown in Figure 19. The greatest displacement is detected on the fragment with the posterior facet (approx. 0.35 mm).

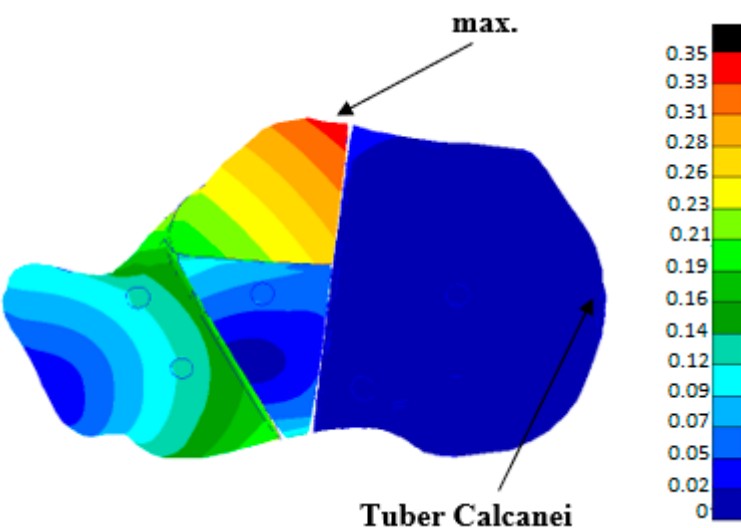

**Figure 19.** Total displacements on the broken heel bone/mm/after C-NAIL application.

Figure 20 shows the total displacements on the calcaneal nail. The size of the greatest displacement was about 0.146 mm. The overview of the calculated values of stresses and displacements is presented in Table 1.

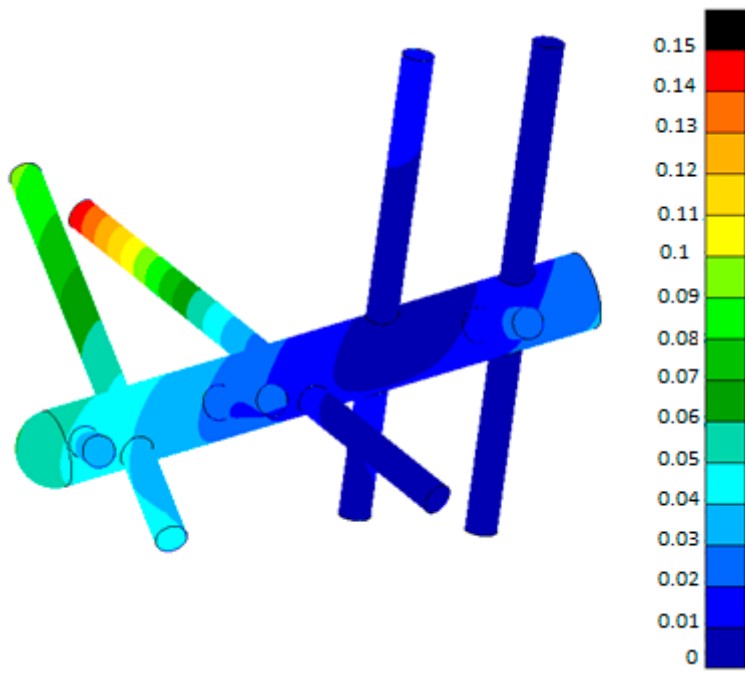

**Figure 20.** The total displacements on the calcaneal nail C-NAIL in the fragmented heel bone/mm/.

**Table 1.** Results of the displacements and stress on the C-NAIL calcaneal nail.

| C-NAIL Displacement/mm/ | Calculated Stress $\sigma_{HMH}$/MPa/(C-NAIL) | Yield Stress (TiAl4V)/MPa/ |
| --- | --- | --- |
| 0.146 | 390.9 | 912 |

## 5. Conclusions

Thanks to the development of implants, it is now possible to treat even very severe and complicated fractures more effectively than in the past. The use of numerical models of bone tissue as well as the osteosynthetic material helps in developing and optimizing

implants thanks to more accurate information about the load distribution on the given implant. Such improved reliability of implants brings the desired improvement in the quality of medical care. Models created in the Materialise Mimics software can also be used to develop medical instruments and implants.

Based on the numerical strength analyses, the authors recommend that the strength of the calcaneal nail C-NAIL as well as the stabilization of bone fragments resulting from its use is sufficient for clinical practice. Similar results were obtained from physical experiments performed in Germany, see [22], as well as in finite element models where the interaction of the calcaneus and the C-NAIL was simulated by an elastic foundation [25].

In the future, the modelling of C-NAIL can be improved through explicitly dynamic analyses, application of probabilistic approaches, enhanced materials and new experiments, e.g., [26–33].

**Author Contributions:** Conceptualization, F.S. and K.F.; Project administration, L.P.; Resources, F.S., K.F. and M.P.; Writing—original draft, F.S. and K.F.; Writing—review & editing, J.H., M.S., Z.M., P.K., M.H., R.M., J.P., O.U. and K.D. All authors have read and agreed to the published version of the manuscript.

**Funding:** This article was supported by international projects CZ.02.1.01/0.0/0.0/17_049/0008441 "Innovative Therapeutic Methods of Musculoskeletal System in Accident Surgery" and CZ.02.1.01/0.0/17_049/0008407 "Innovative and additive manufacturing technology-new technological solutions for 3D printing of metals and composite materials" within the Operational Programme Research, Development and Education financed by the European Union and from the state budget of the Czech Republic and SP2022/26 Computational and experimental modelling in the tasks of applied mechanics and biomechanics.

**Data Availability Statement:** Not applicable.

**Conflicts of Interest:** The authors declare no conflict of interest.

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
