# Peer review of "Numerical Analysis of the Calcaneal Nail C-NAIL"

_applsci, doi:10.3390/app12105265_

Round 1

Reviewer 1 Report

  • The paper investigated the calcaneal nail C-NAIL experimentally and numerically.
  • The finite element method is used for the strength, reliability and safety of the treatment.
  • Finite element mesh seems to be a sufficient approximation to the real bone.
  • Models are created in the Materialize Mimic software.
  • Numerical analysis is performed on the model of heel bone virtually cut into 7 segments.
  • This treatment reduces stress by passing a screw through the fragments of the heels.
  • This research does not inform about which age of people are eligible for this treatment.
  • This study does not satisfy that this treatment will be the same for elders or younger ones.
  • The cutting method is not described fully (only given in the references).
  • The material and type of the screw is not defined which will not cause an infection in the body.

Author Response

Hello,

thank you for your time and support, please see a file with my answers.

Best regards

František Sejda

Reviewer 2 Report

To improve the quality of results I would recommend adding figures with the sectional geometry to make it more clear to the audience where the displacements and stresses have been measured.

Although I am not an English native I would also recommend checking the final version of the native English, as there are several words that I have seen in some scientific context for the first time.

Author Response

(The authors gave the same response as above.)
